# Sequential Rib Labeling and Segmentation in Chest X-Ray using Mask R-CNN

**Jöran Wessel**[1,2]

[1] *Philips Research, Hamburg, Germany*

[2] *Institute of Medical Informatics, Universität zu Lübeck, Lübeck, Germany*

**Mattias P. Heinrich**[2]

**Jens von Berg**[1]

**Astrid Franz**[1]

**Axel Saalbach**[1]

## Abstract

Mask R-CNN is a state-of-the-art network architecture for the detection and segmentation of object instances in the computer vision domain. In this contribution, it is used to localize, label and segment individual ribs in anterior-posterior chest X-ray images.

For this purpose, several extensions have been made to the original architecture, in order to address the specific challenges of this application. This includes the use of rib specific networks, facilitating dedicated anchor boxes sampled from a training set, as well as a sequential processing of all ribs. Here, the segmentation result of the upper neighbor rib is used as additional input to the network.

This approach is the first addressing both rib segmentation and anatomical labeling in chest radiographs. The results are comparable or even better than existing methods aiming only at segmentation.

**Keywords:** Rib Detection, Rib Segmentation, Mask R-CNN, X-ray Images

## 1. Introduction

The identification of ribs has many applications in chest radiography. Ribs may obscure important findings in the lung parenchyma, why rib shadows can be either excluded from the automatic analysis (Candemir et al., 2016) or suppressed from the image (von Berg et al., 2016) to minimize their impact. In contrast to rib segmentation, rib labeling is required in chest X-ray quality assessment. Here, counting the ribs visible in the lung field, is a standard procedure to assure proper inhalation state (Mader et al., 2018). But also other applications like automatically localizing findings from a report or establishing correspondence between follow up images may benefit from a rib segmentation and labeling method. No method published so far does achieve both rib segmentation and rib labeling.

Mask R-CNN (He et al., 2017) is a convolutional neural network for simultaneous object detection and segmentation (i.e. instance segmentation), developed for real time video processing. In the following, we discuss an extension of the Mask R-CNN algorithm and its application to chest X-ray analysis.

## 2. Data and Method

In this work we use the dataset described in (von Berg et al., 2016) which contains 174 posterior-anterior chest X-ray images. The ribs in this dataset were contoured by hand earlier in order to evaluate a bone suppression method, so that only the visible shadows of the ribs are properly annotated. For each X-ray image, the analysis was restricted to the ribs 1 - 9 (which were consistently visible). For the analysis, the images and annotation masks are down sampled to a pixel spacing of approximately 1 mm. This corresponds to an image size of less than $500 \times 500$ pixel. Additionally, the training image set was augmented by using affine transformations.

For our experiments we extend a publicly available implementation[1], with a ResNet50 (He et al., 2016) and a Feature Pyramid Network (Lin et al., 2017) as a feature extractor. The first extension of our method includes the use of dedicated anchor boxes. Instead of independently recognizing and evaluating Regions of Interest with the Region Proposal Network, the network determines shifts on the basis of anchor boxes which correspond to typical rib locations (without Non-Maximum Suppression). In this experiment, the anchor boxes have been estimated using the Mean Shift algorithm (Comaniciu and Meer, 2002). Therefore, 30 cluster boxes were computed from all ground truth bounding boxes of the entire dataset for all labels. In order to compensate for different sizes of the X-ray images, the ground truth boxes are normalized with respect to the size of the respective image.

Due to the high self-similarity of adjacent ribs, reliable rib labeling is a challenging task. Therefore, as a second extension, a sequential processing scheme is introduced, similar to the method from Lessmann et al. (2019). Here, separate networks are trained for each rib so that the networks always segment and classify the same rib for the shifted cluster boxes. Using a ResNet architecture for X-ray image analysis, the image data is typically replicated across the three input channels (RGB). In this setup however, for ribs 2 to 9, the output of the segmentation of the above rib is used for the third channel. The first two channels are filled with the X-ray image in grey values. The idea of this extension is that the additional information of the upper neighbor rib makes the detection of the current rib more precise and reduces false detections.

## 3. Results

For an evaluation of the proposed architecture, a five-fold cross-validation was performed. In Figure 1 the outcome of the algorithm is depicted for one exemplary case. The high agreement between the ground truth annotations (left panel) and the obtained results (right panel) is also reflected in a quantitative evaluation for the predicted bounding boxes and the segmentations.

The results for the detection of the bounding boxes, as well as for the instance segmentation of the ribs are given in Table 1. With the proposed architecture, mean Dice values of 0.846 for the bounding boxes and 0.733 for the segmentation are achieved Compared to the original implementation of the algorithm, this is an improvement of more than 5% and 23%, respectively.

---

1. https://github.com/multimodallearning/pytorch-mask-rcnn

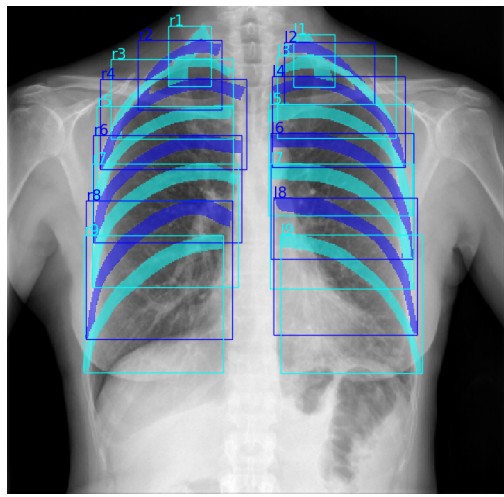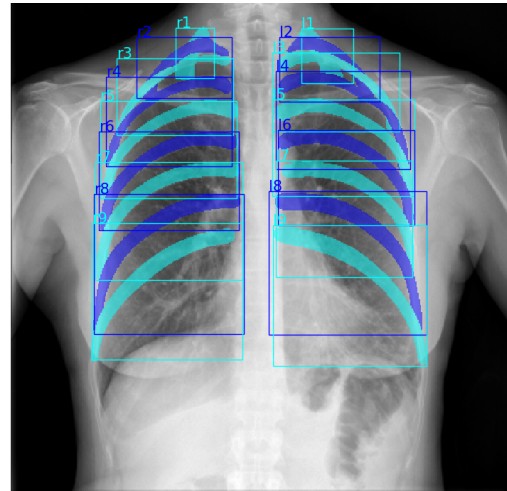

Figure 1: Application of the adapted Mask RCNN algorithm to a chest X-ray image. The ground truth is shown on the left and the prediction on the right. For illustration purposes, next to the segmentation, also the bounding boxes and the rib labels are given.

Table 1: Dice coefficients for rib detection and rib instance segmentation in a five-fold cross-validation. The dataset covers 174 posterior-anterior chest X-ray images (von Berg et al., 2016)

|  | **Detection** | **Segmentation** |
|---|---|---|
| Left | $0.841 \pm 0.126$ | $0.732 \pm 0.207$ |
| Right | $0.850 \pm 0.104$ | $0.734 \pm 0.211$ |

Previously, the problem of rib detection has been addressed by Candemir et al. (Candemir et al., 2016), using atlas-based methods, using accuracy, sensitivity and specificity as evaluation criteria. With values of 0.95, 0.82 and 0.98 compared to 0.86, 0.75 and 0.92, our algorithm achieves better results in all categories.

## 4. Conclusion

In this contribution we presented the first approach for simultaneous rib detection and segmentation in thoracic radiography. The introduced enhancements result in a considerable improved detection rate and segmentation accuracy compared to the original Mask R-CNN approach. Furthermore, it outperforms existing state-of-the-art techniques using atlas-based registration, while providing very fast run-times that enable realtime analysis.

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
