# OpenReview forum: "Sequential Rib Labeling and Segmentation in Chest X-Ray using Mask R-CNN"
_MIDL.io/2019/Conference/Abstract — MIDL Abstract 2019_

### Official Review · AnonReviewer1 · 2019-05-01
**Use of R-CNN for segmentation and labeling of ribs in Chest Xray**

**Rating:** 2
**Confidence:** 2

**Review:**


This paper presents a novel application to identify and segment ribs in chest x-ray data.

Pro
- joint identification and segmentation with appropriate performance


Cons:
- overall limited novelty
- comparison to other methods with performance measures from the literature on different datasets. That's not fair and rather non-sensical.
-clinical application/need for automatic rib labeling is unclear

---

### Official Review · AnonReviewer2 · 2019-05-03
**An approach for instance segmentations of the ribs combining mask-rcnn and sequential networks**

**Rating:** 3
**Confidence:** 2

**Review:**

The authors proposed an adaptation of the Mask R-CNN (He et al. 2017) and the sequential instance segmentation of  Lessmann et al. (2019) for simultaneous rib segmentation and localization in chest X-ray.

Overall the approach seems sound although given the relative complexity of the approach, the abstract only provide a rough overview of it. As such, it is difficult to understand their methodological adaptations with respect to the original Mask R-CNN (beyond the change of CNN architecture).

In terms of results. For this particular application, it is not clear to me what the benefit of estimating the localisation independently of the segmentation is. Providing results from a straightforward multi-class segmentation network would be highly valuable.

The authors claim to achieve a large improvement compared to a previous method  but it is unclear how fair the comparison is. Also, the authors claim they are real time but do not report the duration of the inference for an image and do not explain why this is of interest for their application.

Despite these limitation, the paper may lead to interesting discussion at the conference which is why I am leaning towards acceptance.

---

### Decision · Program_Chairs · 2019-05-06
**Acceptance Decision**

Accept